# Infection with multiple HIV-1 founder variants is associated with lower viral replicative capacity, faster CD4+ T cell decline and increased immune activation during acute infection

**Gladys N. Macharia**[1,2], **Ling Yue**[3], **Ecco Staller**[1,2], **Dario Dilernia**[3], **Daniel Wilkins**[3], **Heeyah Song**[3], **Edward McGowan**[1,2], **Deborah King**[1,2], **Pat Fast**[4], **Nesrina Imami**[1], **Matthew A. Price**[4,5], **Eduard J. Sanders**[6,7], **Eric Hunter**[3,8], **Jill Gilmour**[1,2] *

1 Department of Medicine, Imperial College London, London, United Kingdom, 2 IAVI Human Immunology Laboratory, London, United Kingdom, 3 Emory Vaccine Centre, Yerkes National Primate Research Centre, Emory University, Atlanta, GA, United States of America, 4 IAVI, New York, NY, United States of America, 5 Department of Epidemiology and Biostatistics, University of California at San Francisco, San Francisco, CA, United States of America, 6 Kenya Medical Research Institute-Wellcome Trust, Kilifi, Kenya, 7 Nuffield Department of Clinical Medicine, Centre for Clinical Vaccinology and Tropical Medicine, University of Oxford, Headington, United Kingdom, 8 Department of Pathology and Laboratory Medicine, Emory University, Atlanta, GA, United States of America

* J.Gilmour@imperial.ac.uk

## Abstract

HIV-1 transmission is associated with a severe bottleneck in which a limited number of variants from a pool of genetically diverse quasispecies establishes infection. The IAVI protocol C cohort of discordant couples, female sex workers, other heterosexuals and men who have sex with men (MSM) present varying risks of HIV infection, diverse HIV-1 subtypes and represent a unique opportunity to characterize transmitted/founder viruses (TF) where disease outcome is known. To identify the TF, the HIV-1 repertoire of 38 MSM participants' samples was sequenced close to transmission (median 21 days post infection, IQR 18–41) and assessment of multivariant infection done. Patient derived *gag* genes were cloned into an NL4.3 provirus to generate chimeric viruses which were characterized for replicative capacity (RC). Finally, an evaluation of how the TF virus predicted disease progression and modified the immune response at both acute and chronic HIV-1 infection was done. There was higher prevalence of multivariant infection compared with previously described heterosexual cohorts. A link was identified between multivariant infection and replicative capacity conferred by *gag*, whereby TF *gag* tended to be of lower replicative capacity in multivariant infection (p = 0.02) suggesting an overall lowering of fitness requirements during infection with multiple variants. Notwithstanding, multivariant infection was associated with rapid CD4+ T cell decline and perturbances in the CD4+ T cell and B cell compartments compared to single variant infection, which were reversible upon control of viremia. Strategies aimed at identifying and mitigating multivariant infection could contribute toward improving HIV-1 prognosis and this may involve strategies that tighten the stringency of the transmission

**Data Availability Statement:** All relevant data are within the manuscript and its Supporting Information files.

**Funding:** This work was funded in part by IAVI whose work is made possible by generous support from many donors including: the Bill & Melinda Gates Foundation; the Ministry of Foreign Affairs of Denmark; Irish Aid; the Ministry of Finance of Japan in partnership with The World Bank; the Ministry of Foreign Affairs of the Netherlands; the Norwegian Agency for Development Cooperation (NORAD); the United Kingdom Department for International Development (DFID), and the United States Agency for International Development (USAID). The full list of IAVI donors is available at www.iavi.org. This study was made possible by the generous support of the American people through USAID. The contents are the responsibility of the IAVI and do not necessarily reflect the views of USAID or the United States Government. EJS receives research funding from IAVI, NIH (grant R01AI124968), and the Wellcome Trust. This work was also supported through the Sub-Saharan African Network for TB/HIV Research Excellence (SANTHE), a DELTAS Africa Initiative [grant # DEL-15-006]. The DELTAS Africa Initiative is an independent funding scheme of the African Academy of Sciences (AAS)'s Alliance for Accelerating Excellence in Science in Africa (AESA) and supported by the New Partnership for Africa's Development Planning and Coordinating Agency (NEPAD Agency) with funding from the Wellcome Trust [grant # 107752/Z/15/Z] and the UK government. The funders had no role in study design, data collection and analysis, decision to publish, or preparation of the manuscript.

**Competing interests:** The authors have declared that no competing interests exist.

bottleneck such as treatment of STI. Furthermore, the sequences and chimeric viruses help with TF based experimental vaccine immunogen design and can be used in functional assays to probe effective immune responses against TF.

## Author summary

The development of a safe and effective HIV-1 vaccine alongside cure strategies is a major public health concern and requires in-depth knowledge of the HIV-1 populations that establish infection. Here we sequenced the HIV genetic populations present during the acute stage of HIV-1 infection in 38 Men who have sex with men (MSM) and identified the transmitted/founder variant's sequence. We then cloned the *gag* gene from each patient's transmitted/founder *gag* sequences in both single and multiple variant infection into a common, lab-adapted virus and measured the replicative capacity it conferred on this virus. Finally, cellular immune responses were compared between single variant and multiple variant infection at 0–3, 6–9 and 24–30 months after infection. We observed a higher frequency of multivariant infection in MSM than has been previously described in heterosexually infected participants, and this was associated with faster decline of $CD4^+$ T cells and perturbances in the $CD4^+$ T cell and the B cell compartment. Moreover, the replicative capacity conferred by *gag* was lower in multivariant infection, suggesting a less stringent transmission bottleneck that allowed for less fit variants to establish infection. These results suggest that strategies to tighten the stringency of the transmission bottleneck may be of benefit to patients by reducing the likelihood of multivariant transmission and potentially slowing down disease progression.

## Introduction

HIV infection is now considered a life-long illness that remains a chronic disease manageable only by successfully suppressive combination antiretroviral therapy (cART) but for which no cure or vaccines are available [1]. While advances in treatment methods have resulted in a decline of AIDS-related deaths, only 60% of people infected with HIV-1 know their status and not all those who know their status have access to cART, resulting in 1.4 to 2.3 million new HIV infections every year [1]. Thus, there is still a great need for safe, effective, and scalable strategies to prevent and cure HIV-1 infections worldwide. In order to inform the rational design of new interventions it is critical to understand the complex and dynamic events occurring during acute HIV-1 infection which set the stage for disease trajectory. The outcome of these early events is shaped by both the virus characteristics and multiple host factors which when favorably aligned result in initial control of viremia and preservation of $CD4^+$ T cell counts, albeit to different levels in different patients [2–4]. However, variations in disease progression have been reported despite the presence of favorable host genetic factors including HLA types that are associated with viral control, underscoring the importance of viral characteristics in influencing disease trajectory [5, 6].

Furthermore, HIV-1 mucosal transmission is associated with a stringent and non-stochastic genetic population bottleneck in which a very limited number of HIV-1 variants from a pool of genetically diverse quasi-species in the transmitting partner establish infection [7–9]. Genetic comparisons of transmitted/founder (TF) viruses showed that certain genetic and phenotypic traits which can facilitate viral transmission are selected for, including shorter

variable loops and fewer potential N-linked glycosylation sites for subtype A and C, and some signatures in *env* [7, 10–12]. Phenotypic comparisons between the TF and chronic viruses also demonstrates that TF viruses incorporate higher Env content per viral particle and bind dendritic cells more efficiently but do not necessarily have enhanced infectivity or increased resistance to inhibition by IFN-α than chronic viruses [13, 14]. An accurate characterization of TF virus characteristics is crucial to understanding the role of virus characteristics in HIV-1 pathogenesis and shaping disease progression, and in identifying target immunogens and novel vaccine strategies.

Despite the severe population bottleneck, around 20% of mucosal HIV-1 transmissions are still established by multiple HIV-1 TF viruses [15, 16]. The role of the route of transmission in susceptibility to HIV-1 infection has been previously studied and rectal and percutaneous inoculation both shown to be associated with increased odds of transmission and therefore increased likelihood of multivariant infection [15, 17, 18]. The mechanisms for transmission of multiple HIV-1 variants into the same subject is still unknown. It has been hypothesized that during transmission of multiple HIV-1 variants, either the transmission rate is transiently increased due to certain cofactors, such as inflammatory genital infections that may reduce the transmission barrier by destroying the intact mucosa or increasing the availability of activated CD4$^+$ T cells, or that transmission of multiple variants are linked events, such as through a multiply-infected cell or via viral aggregates [19, 20]. It is therefore also not known whether variants transmitted to the same HIV-1-infected individual share similar phenotypic properties or are a result of a less stringent transmission bottleneck that allows any reasonably fit virus to establish infection. Previous analysis of infection established by multiple TF variants shows that the frequency of individual variants in multivariant infection can be quite fluid, with variants increasing and decreasing in frequency during early infection in some individuals [20]. A role for multi-variant infection in HIV-1 pathogenesis or disease progression in natural HIV-1 cases has been implicated in previous studies [21]. Studies of the HIV-1 breakthrough infections in the Step and RV144 HIV-1 vaccine efficacy trials showed that although recently infected individuals with multiple phylogenetically linked HIV-1 founder variants represented a minority of HIV-1 infections, more diverse HIV-1 virus populations in early infection were associated with significantly higher viral load 1 year after HIV-1 diagnosis [22]. In both trials, heterogeneity of founder viruses was also identified as a viral determinant of a more rapid CD4$^+$ T cell decline.

As the TF evolves to adapt to the newly infected patient, escape within *gag* and *pol* encoded epitopes can significantly reduce viral replicative capacity and has been associated with lower viral loads following transmission [23, 24]. Insertion of the *gag* gene from the TF into a proviral vector has shown that certain mutations in the transmitted *gag* confer an increased replicative capacity while some reduced the virus' replicative capacity independent of HLA type [25–27]. In addition, individuals infected with attenuated *gag* sequences are delayed in their progression to CD4$^+$ T cell counts <300 cells/μl of blood compared to high replicating viruses. They also have lower levels of activated CD8$^+$ T cells and proinflammatory cytokines indicating a long-term clinical benefit associated with the transmission of less fit viruses and positioning aberrant immune activation as a link between viral fitness and disease progression [25].

The Kilifi Protocol C cohort of longitudinally followed MSM with incident HIV infection presents a unique opportunity to characterize TF viruses in a group where disease outcome is known [28]. The aim of the present study was to characterize the founder variants and describe the role of multivariant infection in HIV pathogenesis in these MSM participants. Single genome amplification was performed on plasma collected during acute infection and the sequence of the TF virus(es) inferred from multiple amplicons. Chimeric viruses containing the TF *gag* gene were measured for replicative capacity in both single variant and multivariant

infection. This allowed us to investigate the relationship between viral fitness, the transmission bottleneck, and HIV pathogenesis in the context of multivariant infection.

## Results

### Frequency of multivariant HIV-1 infection in MSM

From 38 participants, a total of 120 near full-length genome amplicons, and 537 half genome amplicons were generated. Both full-length and half genome amplicons were sequenced using single-molecule long-read sequencing and MDPSeq analysis pipeline (Dilernia et al., 2016). The near full-length HIV-1 genome contained all the HIV-1 genes but with truncations at the 5'R-U5 and 3'R regions. Specifically, the internal nested 5'primer corresponded to the first 30 nucleotides of U5, and the internal nested 3'primer corresponds to the last 30 nucleotides of the 3'R region, thus rendering these two short stretches of 30 nucleotides primer derived rather than authentic virus sequences. Although the 5' and 3' half genomes overlapped by the entire *vif* gene, they were treated as independent half genomes and not assembled into near full-length genomes due to the extreme homogeneity within *vif* sequences early in infection. A phylogenetic tree of all 3' and 5' half-genome sequences is presented in S1A **and S1B** Fig and confirms phylogenetic relatedness of the intra-patient quasispecies.

There was remarkable within-patient homogeneity of sequences, with most genomes differing by less than 10 nucleotides across the entire ~9kb HIV-1 genome. The median number of sequences analyzed per patient was 15 (range, 6–42) with the within-patient diversity ranging from 0.05% to 0.3%. In all cases, nucleotide changes from the consensus sequence in each amplicon were randomly spread across the whole genome and the sequences that corresponded to the consensus sequence was inferred to be the founder virus sequence. (**Fig 1A and 1B**).

For each subject, the frequency distribution of all intersequence average pairwise distances defined as the average number of base positions by which intra-patient genomes differed was determined. Two measures of diversity were used: a categorical measure that distinguishes between subjects with a single founder variant or multiple founder variants and a continuous measure of diversity which corresponds to the mean pairwise diversity among sequences from a subject. An infection was classified as occurring from a single founder variant when all HIV-1 sequences from the recipient formed a monophyletic clade with boot-strap support >70 percent (**S1A and S1B Fig**). Using this approach, 39% (15 out of 38) of these MSM volunteers were infected with multiple founder viruses (median 2, IQR 2–4, range 2–12), much higher than has been described for heterosexual transmission [7, 11, 12] but consistent with observations in other MSM cohorts [7, 18, 29, 30]. Each lineage contained a unique set of identical or near identical sequences suggesting that these subjects were infected by very closely related viruses (**Fig 1A**), most likely from one partner as opposed to a single virus that evolved into two distinct lineages in the brief period preceding peak viremia, or superinfection. Multivariant transmission was evident by the phylogenetic tree topology and highlighter plot alignments (**Fig 1B**) and by the higher pairwise distances in both the 3 prime (p = 0003) and 5 prime half genomes (p<0.0001) (**Fig 1C**). There was no evidence for increased diversity in variants that were sequences at the earliest timepoint (~14 days post infection) compared to those that were sequences later during acute infection (~60 days post infection) (**Fig 1D**), suggesting that the sequences were obtained prior to viral adaptation to the new host and were therefore close to the initial TF(s).

### Multivariant infection was associated with faster decline of CD4 T cells

An investigation of whether infection with multiple founder variants resulted in different disease outcome using CD4$^+$ T cell count and viral load measurements as measures of

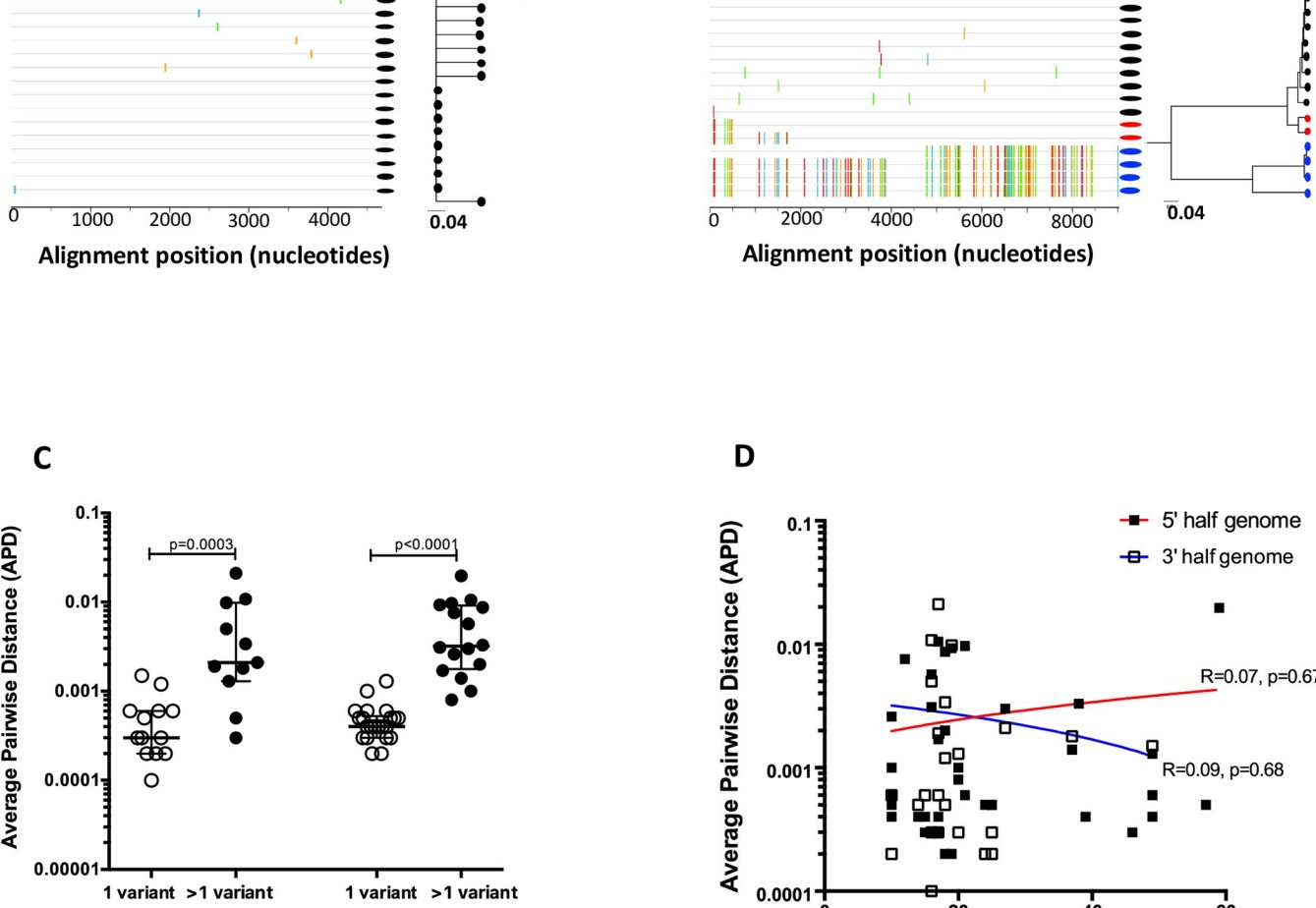

**Fig 1.** A and B) Highlighter plots and ML branches of single variant and multivariant infection respectively. Two distinct variants are denoted in black and blue while inter-variant recombination is denoted in red on Fig 1B. C) Average pairwise distance (APD) of HIV-1 SGA/S sequences distinguishes between single and multiple founder variants. D) Average pairwise distance (APD) plotted against days post infection showing no evidence for increased diversification and confirming that samples in this time window were close to the wild type transmitted/founder variant.

disease outcome was done. Infection with multiple founder viruses resulted in lower CD4$^+$ T cell counts in the first 5 years of infection with patients infected with multiple founder viruses reaching CD4$^+$ T cell counts under 350 cells/mm$^3$ faster than those infected with a single variant (**Fig 2A**, p = 0.001) but viral load set points were similar regardless of the multiplicity of founder variants (**Fig 2B**, p = 0.4). These results suggest that greater heterogeneity in the HIV-1 founder population of recently infected individuals is associated with a worse disease outcome in a manner that is independent of virus load in blood within the timepoints tested and further provides evidence that pathogenic events at acute HIV-1 infection disproportionately contribute to disease progression even after control of viremia.

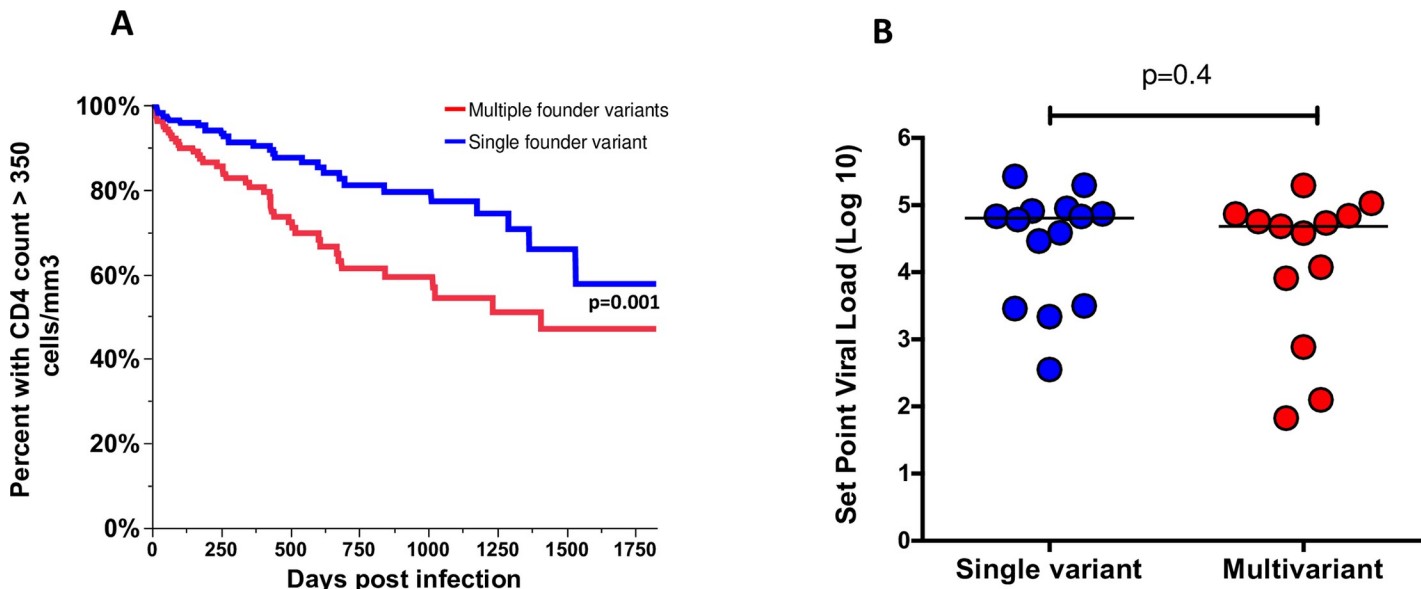

**Fig 2.** (A) Infection with multiple founder viruses resulted in lower CD4+ T cell counts in the first 5 years of infection with patients infected with multiple founder viruses reaching CD4+ T cell counts under 350 cells/mm³ faster than those infected with a single variant. B) There were no differences in set point viral load.

## Frequency of variant in multivariant infections was not driven by replicative capacity

In the infections in which multivariant infection was detected, the proportional representation of the variant within that patient's sampled HIV-1 population was examined by estimating the relative frequencies of each variant within the viral population. Variants were conservatively defined as minority variants when they represented <10 percent of the observed viral population. The proportions of TF viruses were highly variable, with major TF viruses as high as 96% and minor TF viruses as low as 3% (**Fig 3A**). These highly unequal proportions of TF viruses at acute infection suggest different replication advantages among TF viruses in the same HIV-1-infected individuals. We therefore measured the replicative capacity conferred by TF *gag* sequences of both major and minor variants, which has previously been shown to correlate closely to that of the entire full length infectious molecular clone, and to be predictive of disease progression [26, 31, 32].

We performed in depth analysis on 7 out of the 15 cases of multivariant infection from whom more than one infectious *gag*-NL4.3 chimera was cloned, and 5 out of the 15 cases of multivariant infection from which we were able to clone at least one variant. Several clones that did not yield virus titers that were sufficiently high for use in our format of the viral replicative capacity assay, including all the variants from 3 out of the 15 cases of multivariant infection were not analyzed. Insertion of patient derived *gag* genes on to NL4.3 altered the replicative capacity of NL4.3, with a wide range of replicative capacity of intra patient founder viruses and no evidence for selection of similar replicative capacity within each patient (**Fig 3B**), suggesting that in cases of multivariant infection, any reasonably fit virus could be transmitted. However, the major variant (termed TF1) often displayed lower replicative capacity than the minor variants (TF2 to TF5). We therefore performed a matched pairwise comparison between the replicative capacity of the major variant and the mean replicative capacity of the minor variant(s) in the 7 cases where more than one variant had been cloned and observed lower replicative capacity in the major variants compared to the minor variant (Wilcoxon

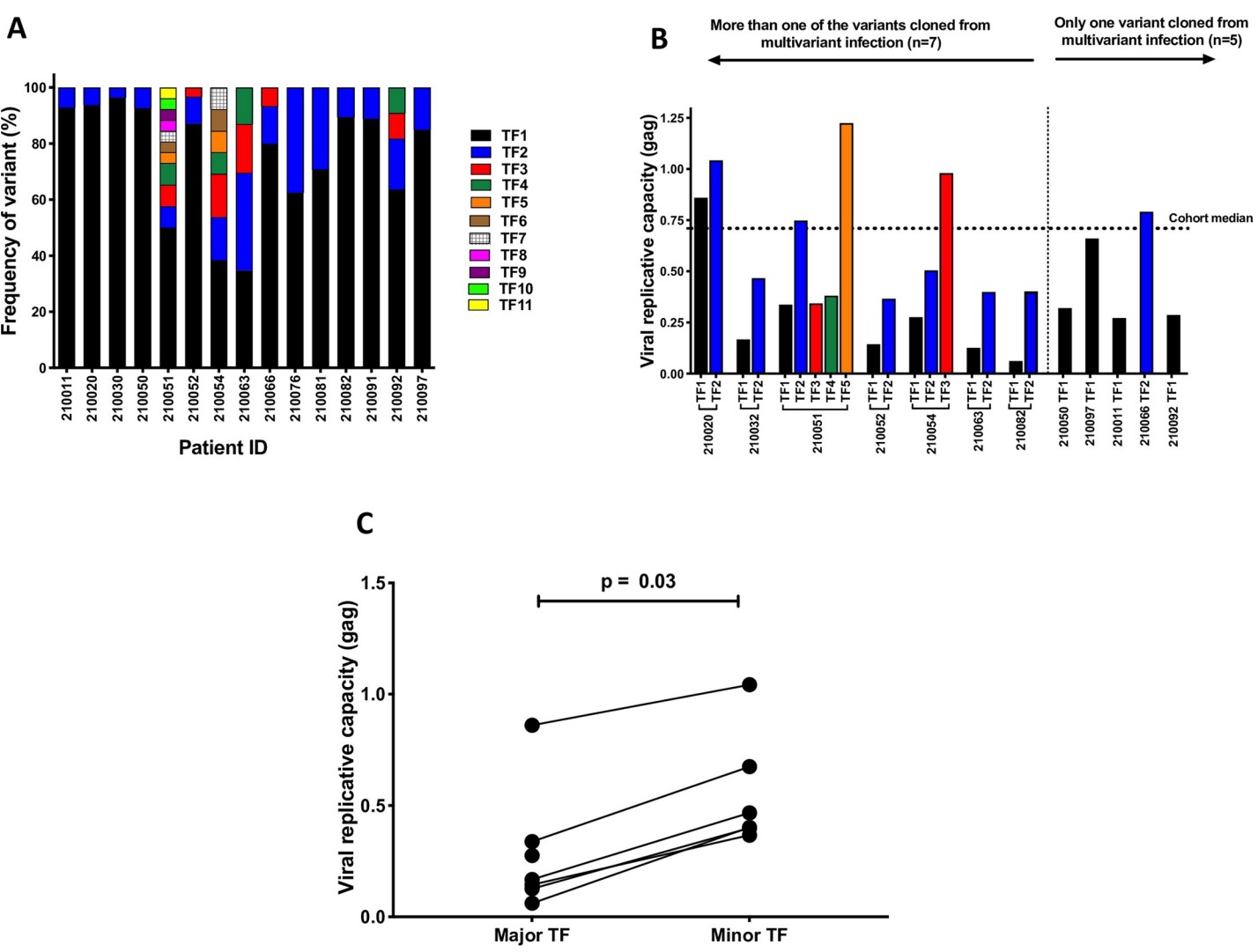

**Fig 3.** A) Proportion representation of multiple transmitted/founder viruses showing unequal representation of variants, with major variants representing as much as 96% of the virus populations and minor variants representing as low as 3% of the virus population sampled. B) A wide range of replicative capacity of intra patient founder viruses was observed, with no evidence for selection of similar replicative capacity. More than one TF *gag* variant was cloned and tested in 7 participants and only one variant was tested in 5 participants. The major variant (termed TF1) often displayed lower replicative capacity than the minor variants (TF2 to TF5). C) Matched pairwise comparison between the replicative capacity of the major variant and the mean replicative capacity of the minor variant(s) in the 7 cases where more than one variant had been cloned showing lower replicative capacity in the major variants compared to the minor variant (Wilcoxon matched pairs signed rank, p = 0.03).

matched pairs signed rank p = 0.03, Fig 3C). This suggested that the overrepresentation of some sequences within the patient were not as a result of a fitter *gag* gene and may have been a result of either a high inoculation dose of that variant, or a higher fitness of genes outside of *gag*.

## Multivariant infection was characterized by lower replicative capacity and associated with recent *Neisseria gonnorhoeae* infection

It is not clear whether viral replicative capacity plays a role in the HIV-1 transmission process. Some studies into transmission pairs have suggested that founder viruses possess higher replicative fitness than non-transmitted variants, but other studies find no evidence for increased

fitness or particle infectivity [33, 34]. Moreover, the host factors that select for founder viral replicative capacity or allow for propagation of one variant over the other are not well understood and may operate in the local mucosa or local lymph nodes. The replicative capacity of founder variants derived from multivariant infection (n = 12, mean RC where more than one variant was characterized) was therefore compared to that of variants derived from single variant infection (n = 18) to elucidate whether there was a relationship between the ability of a variant to establish infection and its replicative capacity. Even with our small sample size, founder viruses of multivariant infection exhibited lower median replicative capacity than did those of single variant infection (p = 0.02, **Fig 4A**). We interpret this to suggest that in the transmission of multiple HIV-1 variants, the fitness requirements for transmission may be transiently lowered and transmission rate transiently increased due to certain cofactors in the patients, such as inflammatory genital infections that may reduce the transmission barrier by destroying the intact mucosa or increasing the availability of activated CD4+ T cells, and therefore reduce the fitness requirements for transmission.

A subset of the 38 MSM (n = 15) with symptoms compatible with a sexually transmitted infection were assessed for Gonococcal infection. Neisseria gonorrhoeae detected in the 6 months leading up to HIV infection, which has previously been shown to strongly predict HIV acquisition in this cohort [35, 36]. A total of 5 out of the 15 MSM had gram stain confirmed gonococcal infection in the 6 months prior to HIV-1 infection. Four of these 5 confirmed *Neisseria gonorrhoeae* cases had multiple HIV founder variants while only one out of five had a single HIV founder variant (Fig 4B). Overall, 57% of the volunteers infected with multiple transmitted/founder variants and tested for *Neisseria gonorrhoea* (4 out of 7) had a

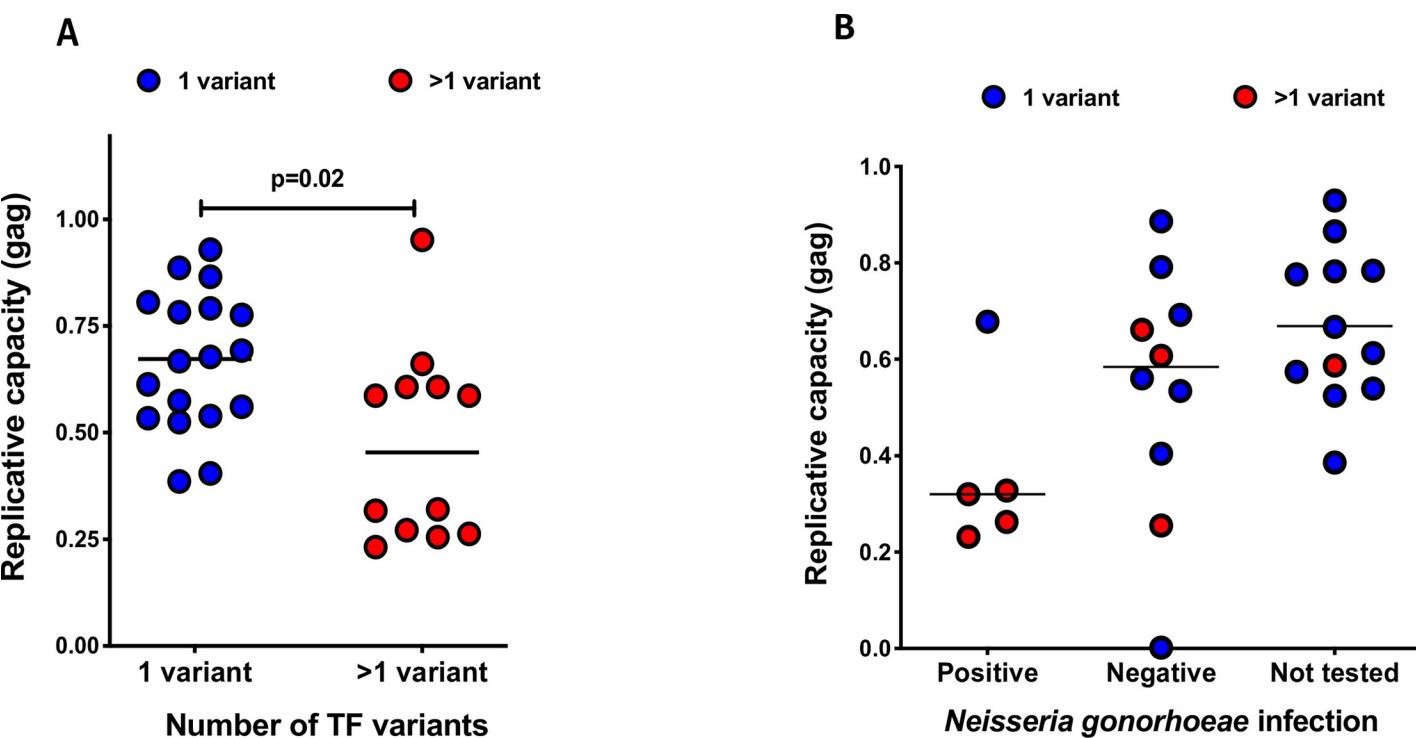

**Fig 4.** A) Multivariant infection was characterized by lower replicative capacity, suggesting that multiple founder variant infection resulted from a less stringent transmission selection process. Replicative capacity values in multivariant infection represent the median replicative capacity of all variant tested for each individual. B) Graphical representation of the distribution of participants who had lab confirmed *Neisseria gonorrhoeae* in relation to acquisition of multiple transmitted founder variants and the replicative capacity of those variants. 4 out of 5 confirmed *Neisseria gonorrhoeae* cases accompanied multivariant HIV infection.

recent *Neisseria gonorrhoeae* infection compared to 12.5% (1 out of 8) of those who were infected with a single transmitted/founder variant and tested for *Neisseria gonorrhoea*. Despite the small sample size, these findings align with our previous hypothesis that a breach of the genital mucosa resulting from recent STI infection is a risk factor for the acquisition of multiple transmitted/founder variants and should be investigated in a larger sample size.

## Multivariant infection was associated with immune perturbation during acute infection

One of the hallmarks of HIV-1 infection is chronic activation of the immune system that not only increases the number of activated CD4$^+$ target cells but also directly impairs the immune system through activation-induced cellular exhaustion [37–39]. Such chronic immune activation is established early and often persists following cART and is a more reliable predictor of disease progression than viral load [40, 41]. Therefore, we assessed whether multivariant infection plays a role in driving immune activation by comparing levels of cellular activation in cases of multivariant infection to single variant infection at three time points; 0–3, 6–9 and 24–30 months after infection. Multivariant infection was associated with multiple perturbations of the CD4$^+$ T cell compartment at 0–3 months post infection, with CD4$^+$ T cell phenotypes appearing similar between the two groups at all other time points. During acute infection, patients infected with multiple TF had lower frequencies of effector memory CD4$^+$ T cells (p = 0.01, **Fig 5A**) and transitional memory CD4$^+$ T cells (p = 0.01, **Fig 5A**). In addition to cellular immune activation, T-cell exhaustion is characteristic of pathogenic HIV/simian immunodeficiency virus (HIV/SIV) infection [42, 43]. Exhaustion of T cells is marked by the increased expression of the inhibitory receptor, programmed death 1 (PD-1), and levels of PD-1 expression predict the rate of disease progression [44]. We observed a trend towards increased frequencies of PD1 expressing central memory CD4$^+$ T cells during acute infection (p = 0.5, **Fig 5A**) suggesting early exhaustion that was reversible after the acute infection stage.

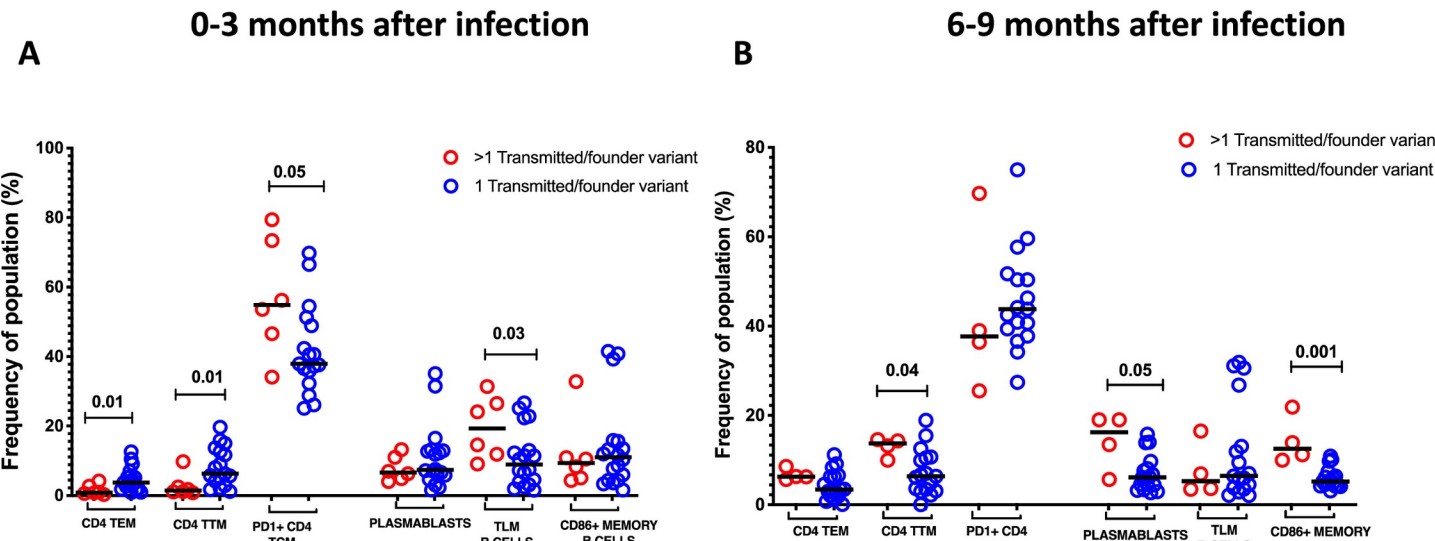

**Fig 5. A)** Infection with more than one transmitted/founder variant was associated with lower frequencies of CD4 TEM (Terminal Effector CD4$^+$ T cells), CD4 TTM (Transitional CD4$^+$ T cells), higher frequencies of exhausted CD4 TCM (PD1+ Central Memory CD4+ T cells) and TLM (Tissue like memory) B cells at 0–3 months after infection. **B)** At 6–9 months after infection, there were higher frequencies of CD4 TTM (Transitional CD4$^+$ T cells), but increased frequency of plasmablasts and CD86 expressing memory B cells.

Unlike in CD4$^+$ T cells, perturbation to the B cell compartment were observed at both 0–3 months and 6–9 months, although not at 24–30 months after infection. Patients infected with multiple founder variants had higher frequencies of tissue like memory B cells at 0–3 months after infection (p = 0.03, **Fig 5A**), a phenotype which has been previously observed during acute HIV infection and associated with viremia [45], but not previously linked to any viral phenotype or to multivariant infection. At 6–9 months, multivariant infection was characterized by increased frequency of plasmablasts (p = 0.05, **Fig 5B**) which in chronic HIV infection mainly produce autoreactive and polyreactive antibodies [45, 46], and an increase in activated B cells that express the activation marker CD86 (p<0.001, **Fig 5B**).

## Discussion

A primary goal of HIV vaccine development is to prevent acquisition of HIV-1 at mucosal surfaces. As HIV-1 rapidly evolves following transmission, it is crucial for pathogenesis studies to elucidate the properties of viruses collected soon after transmission, prior to extensive adaptation in the new host. Our results here build on current understanding of the HIV transmission bottleneck by providing evidence that founder viruses in cases of a less stringent transmission bottleneck as happens in multivariant infection are of lower replicative capacity, but nonetheless multivariant infection is associated with faster decline of CD4$^+$ T cells. Our findings advance previous understanding of the nature of the HIV-1 transmission bottleneck, and the consequences of a less stringent bottleneck.

Consistent with other studies, infection in MSM was associated with increased multiplicity of founder viruses than has been previously shown in heterosexual cohorts [7, 18, 19, 29, 30]. Although we observed a high number of multivariant infections, a majority of patients had a single or very few viruses transmitted and this is consistent with epidemiological observations of the relative inefficiency of virus transmission by most sexual routes [47–49]. Receptive anal intercourse has a reported hazard of 10 times that of penile-vaginal intercourse [50]. The MSM in this study also have high risk behavior as evidenced by documented prevalence of inflammatory STI [35, 51–53] and multiple partners [54, 55] coupled with a social healthcare system that is not sensitized to accurate diagnosis and treatment of STI amongst MSM [56, 57]. STI testing in this cohort was targeting symptomatic individuals and had shown that *Neisseria gonorrhoea* confirmed by gram staining was the strongest predictor of HIV acquisition increasing the risk of infection by up to 15-fold [35, 36]. Of note relatively high prevalence of asymptomatic STI has been shown in in this population[35, 53]. Our observation that *Neisseria gonorrhoea* infection was more frequent in individuals who were infected with multiple transmitted founder suggests that ulcerative bacterial STI predisposes to multivariant HIV infection. This should be investigated further in a larger sample size. In addition, the probability of HIV transmission per unprotected sexual contact is closely related to the donor's viral load [50, 58–60] and recency of infection with participants with acute HIV infection more likely to transmit multiple transmitted/founder variants [61, 62]Thus, differences in risk behavior, routes of virus transmission (receptive vs insertive anal intercourse), clinical stage of and viral load of the source of infection, and presence of STI in recipient may have also influenced the numbers of viruses transmitted and therefore subsequent disease natural history. These factors when in concert with the well-known vulnerability of the thin simple columnar epithelium that lines the rectal mucosal surface as a site of entry [63, 64] suggest that multivariant infections may be a result of both biological and social disadvantages. Our findings regarding the number of viruses leading to productive clinical infection are minimal estimates, and additional viruses could have been transmitted but not sufficiently propagated in vivo to allow detection within the timing of our sampling or our depth of sequencing. Previous findings of

low multiplicity infection and limited viral evolution preceding peak viremia have been interpreted to suggest a crucial but finite window of potential vulnerability of HIV-1 to vaccine-elicited immune responses, but our observation of multiple TF variants suggests that this window may be attenuated in MSM.

Previous studies have shown that in most cases of multivariant infection, the minor variant persists for a long time, often rapidly overtaking and replacing the major variant as a result of an immune response to the major variants and itself then collapsing as it starts to be targeted by CTL responses [20, 65]. Moreover, the viral phenotypic properties that are selected for during multivariant HIV-1 transmission as well as their impact on disease progression remain unclear. The highly unequal proportions of TF viruses at acute infection suggest different replication advantages among TF viruses in the same HIV-1- infected individuals. Some studies have suggested that HIV-1 transmission selects for viruses with high infectivity and replication capacity, and that replicative capacity of the founder virus can predict the rate of subtype C disease progression [25, 26, 32, 33, 66]. Furthermore, previous studies in HIV-1 subtype C had shown that the replicative capacity conferred by *gag* correlates with that of the full-length infectious molecular clone[25, 26], indicating that although other genes play a role in defining in vitro HIV-1 replicative capacity, the contributions of *gag* are a significant component of the replicative capacity of the full-length virus, and *gag* chimeras present an important tool in defining the role of *gag* in the overall viral fitness. We therefore assessed the relationship between replicative capacity conferred by the transmitted *gag* gene which encodes the major structural proteins within the virion sequence and multivariant infection. Paradoxically, we observed that in cases of multivariant infection, the major founder variants did not yield the highest replicative capacity Gag chimeric viruses. Our data suggests that in cases of multivariant infection, any reasonably fit virus could get transmitted, as variants isolated from the same patient often differed in replicative capacity. This is consistent with findings from epidemiologically linked transmission couples that the donor swarm of viruses often comprises of variants of differing replicative capacities, with the variant that was of the highest replicative capacity amongst the donor's swarm not always the variant that establishes infection their newly infected patient [33]. Moreover, the observation that the gag genes derived from the major variants tended to replicate slower than those derived from minor variants suggests that the replication capacity conferred by *gag* may not have been responsible for that variant's dominance over other minor variants within that patient. Alternatively, there may have been increased outgrowth of the major variant either in the mucosa or lymph nodes (and thus not sampled in this study) as a result on fitness conferred by genes outside of *gag* and therefore not captured in our replicative capacity assay. Without mucosal samples taken at the time of transmission, the possibility that the major TF variants were clonally amplified at the donor mucosa at the time of transmission cannot be ruled out, although downstream bottlenecks may reduce or obscure any transmission advantage associated with TF variant frequency in the transmission fluid. Due to the known influence of HLA associated preadaptation to disease progression [67–70], we investigated whether the TF variants in the 7 cases where we could study both major and minor variants carried known HLA escape mutations that were associated with the patients HLA 1 alleles, as this could account for their differences in replicative capacity. However, the escape mutations were present in both the major and the minor variants, suggesting no role in the observed differences in replicative capacity between major and minor variants.

Even with our modest sample size of 38 men, our results were consistent with other studies that showed worse disease outcome in subtype B multivariant HIV infection. We observed that multivariant infection resulted in lower CD4+ T cell counts in the first 5 years of infection with patients infected with multiple founder viruses reaching CD4 T cell counts under 350 cells/mL much faster than those infected with a single variant, and an overall inverse

correlation between the number of founder virus variants and CD4$^+$ T cell counts longitudinally. There was no significant difference in longitudinal viral load measurements over time and viral load set points were similar regardless of the number of infecting founder virus variants, suggesting that greater heterogeneity in the HIV-1 founder population of recently infected individuals is associated with a worse disease outcome in a manner that is independent of antigen load within the timepoints tested. This contrasts with the observation from both the STEP and RV144 HIV-1 vaccine trials as well as a study in subtype B infected American MSM in which multivariant infections was associated with significantly higher viral load one year after HIV-1 diagnosis in both vaccine participants and MSM [22, 29]. However, in these studies HIV sequences were obtained at the time of diagnosis which was not always during acute infection as we did in the current study and involved subtype B infected MSM in the US. The only study that included subtype A infection in an investigation of multivariant HIV infection was on heterosexual transmission and did not investigate disease progression[71]. Increased diversity of viral quasispecies as a result of multivariant infection is likely a double-edged sword; while increased diversity of TF variants might theoretically increase the breadth of the early adaptive immune response which would be expected to lead to better immune mediated viral control, the increased diversity also presents greater opportunity for CTL escape, and it is likely that the balance of these two possibilities plays a role in influencing disease outcome.

The viral sequences in the current study were limited to acute infection and therefore we cannot precisely predict the contribution of each variant to disease progression. Although we were unable to accurately measure peak viral load in this cohort of HIV-1 acutely infected individuals, it seems likely that viruses with high replicative capacity have the potential to induce much higher levels of peak viremia, which then induces exacerbated immunopathology that cannot be completely reversed by the immune response. This data suggests that in the absence of effective cART, factors associated with a less stringent transmission bottleneck resulting in multivariant infections could send the patient on a trajectory of worse disease outcome.

Finally, we identify perturbations in the CD4$^+$ T cell and B cell compartments that are associated with multivariant infection. Interrogating the nature of the immune response in the period before the establishment of a viral set point and at a period while the founder virus is still relatively unevolved to the newly infected patient is paramount to understanding the interplay between virus and host that can be harnessed towards the design of effective interventions to either stop HIV dissemination or bias disease outcome towards a more favorable trajectory. The causes of HIV-1-associated immune activation established in early HIV-1 infection are not clearly defined. Multiple related events contribute to such activation, including direct viral infection of immune cells, pro-inflammatory cytokine production by innate cells (which drives both direct and bystander activation of other immune cells), translocation of microbial products into the blood through damaged intestinal epithelium, loss of virally infected regulatory T cells and chronic mycobacterial and viral co-infections [37, 45, 72–76]. Consistent with all these findings, it has been confirmed that viral load is only an indirect contributor to the rate of progression to AIDS, that immune activation predicts changes in CD4$^+$ T cells, which are stronger and independent of viral load, and that the effect of anti-retroviral therapy in increasing CD4$^+$ T-cell counts better correlates with the decrease in immune activation than the suppression of viral load [38, 75, 77, 78]. The observation that residual immune activation persists despite successfully suppressed viremia highlights the importance of identifying, and then targeting, the mechanisms that cause immune activation.

Our data suggests that multivariant infection plays a role in driving immune dysfunction in both the CD4$^+$ T Cell and B cell compartments. The lower frequencies of multiple important CD4$^+$ T cell subsets and increased exhaustion that we observed in multivariant infection are

consistent with faster depletion of CD4$^+$ T cells. Moreover, increased frequency of tissue like memory B cells that are an exhausted phenotype alongside increased frequency of plasmablasts and the costimulatory receptor CD86 on activated B cells has been previously observed during acute HIV infection and associated with viremia [39, 45, 79], but has not been linked to any viral phenotype or to multivariant infection in the absence of higher viral load. The process of immune perturbance as a result of multivariant infection, however, appears to be reversible post-acute infection, with no differences between patients with single vs multiple variant infection after two years of infection. It is plausible that the observed perturbances in CD4$^+$ T cell and B cell phenotypes may have resulted from the increased STI infection that accompanied multivariant infection, which may then normalize upon STI treatment which was provided for all symptomatic participants. However, most *Neisseria gonorrhoea* infections are subclinical and norfloaxin and ciprofloxacine which were the recommended treatment for urethral or rectal discharge at the time are likely to have been ineffective due to documented resistance [35, 53]. Participants are therefore likely to have had repeated STI [80]. However, consistent with the reports that events at acute infection disproportionately set the stage for disease trajectory [81–83], despite recovering from the observed immune perturbances, patients of multivariant infection still lost their CD4 T cells faster that those infected with a single founder virus.

In conclusion, the current data suggests that strategies to mitigate multivariant infection will be beneficial by slowing down disease progression. In the context of current test and treat guidelines, mitigating perturbances of the immune system that can be difficult to correct even with cART will also be beneficial. These will most likely be similar to strategies aimed at reducing transmission such as the treatment of STIs and increased uptake of PrEP.

## Materials and methods

### Study subjects and ethics statement

Written informed consent was obtained from all study participants and the study was approved by the ethical review board at the Kenya Medical Research Institute (KEMRI).

HIV-1$^+$ samples were drawn from a large well characterized prospective multicenter HIV-1 infection incidence study [28]. Adults aged 18–49 years were eligible if they were HIV-1-sero-negative and reported transactional sex work, a recent STI, multiple sexual partners, sex with an HIV-1-infected partner or unprotected anal sex. Individuals who were seronegative at initial screening but tested positive for HIV-1 p24 at subsequent visits were invited to enroll into an acute infection cohort, where blood samples were taken either monthly or quarterly for HIV-1 plasma viral load and CD4$^+$ T-cell assessments and are the focus of this study. Of the 38 study participants 38 included, 27 were on monthly follow-up and 10 were in quarterly follow up at the HIV positive date. The estimated date of HIV-1 infection (EDI) was calculated as 10 days before the sample collection date when the sample had a positive HIV-1 RNA level and negative p24 antigen and HIV-1 serology, 14 days before a positive p24 antigen test or the mid-point between a previously negative and subsequently positive HIV-1 serologic test, in the absence of either a positive HIV-1 RNA level or p24 antigen test. Patients were included in this study if they had been enrolled before 45 days after the estimated date of infection and were antiretroviral therapy naive. Participants also needed to have a plasma sample available within the first 45 days after the estimated date of infection to allow for definition of the transmitted/founder genome, and PBMC sample at 3, 9 and 24 months after infection to allow for the characterization of immune responses. All samples used in this study were from antiretroviral therapy naive patients with no access to PrEP prior to HIV acquisition. Patients received HIV prevention counselling and when HIV-1 infected were referred for HIV-1 treatment as per the national guideline, which was to initiate treatment at CD4 T cell counts ≤350 cells/mm$^3$

blood. Participants with genital symptoms were provided syndromic treatment, and where suspected *Neisseria gonorrhea* was investigated by gram staining and detection of gram-negative intracellular diplococci consistent with *Neisseria gonorrhea* in urethral or rectal secretions.

The characteristics of the patients involved in this study are summarized in **S1 Table**.

## Amplification and sequencing of near full-length HIV genomes

RNA was extracted from the earliest available plasma, earlier than 45 days after estimated date of infection using the QIAamp Viral RNA Mini Kit (Qiagen, Netherlands) following the manufacturer's instructions. RNA was reverse transcribed using either the SuperScript III or Superscript IV cDNA synthesis kit (Life Technologies, USA) according to the manufacturer's instructions using either primer 1.3'3'PlCb or OFM19 (primer sequence detailed in **Table 1**). Near full length single genome PCR was performed by serially diluting cDNA as previously published [84] to yield a 9-kb fragment beginning at the first nucleotide of the U5 region of the 5'long terminal repeat (LTR) and extending to the last nucleotide of the R region of the 3'LTR. Both rounds of PCR were performed in 1x Q5 Reaction Buffer, 1x Q5 High GC Enhancer, 0.35 mM of each dNTP, 0.5 μM of primers and 0.02 U/μl of Q5 Hot Star High-Fidelity DNA Polymerase (NEB) in a total reaction volume of 25 μl. First round primers were, 1U5Cc and 1.3'3'PlCb, and second round primers were 2U5Cd and 2.3'3'plCb. Cycling conditions for both reactions are 98°C for 30s, followed by 30 cycles of 98°C for 10s, 72°C for 7.5min, with a final extension at 72°C for 10min. Some patients' virus was amplified in two 5kb half genomes overlapping by the Vif gene using the internal primers Vif1 and VifR1 for the first round, and Vif2 and VifR2 for the second round. All products derived from cDNA dilutions yielding <30% PCR positive wells and ~9-kb or ~5-kb in length as appropriate were further amplified using primers 1.3'3'PlCb and 1.U5Cc with a 20-nucleotide barcode added to the primer, and PCR product purified using the Promega PCR clean up kit as per manufacturer's recommendations (Promega, USA).

## SMRT sequencing of HIV-1 genomes

Positive ~9kb single genome amplicons were gel-extracted using the Wizard SV Gel and PCR Clean-Up System (Promega). Genome libraries containing multiple purified HIV-1 genome amplicons were constructed by pooling equal amounts of independently amplified single genomes derived from different patients to a final concentration of 3000ng. SMRTbell libraries were then generated for each pool according to protocols from the DNA Template Prep Kit 2.0 for 10kb amplicon (Pacific Biosciences Inc, California, USA). The quality of the library was assessed by running the sample in the Agilent 2100 Bioanalyzer system (Agilent Technologies,

**Table 1. Primers used for cDNA synthesis and single genome amplification.**

| Primer name | Primer sequence | Primer use |
|---|---|---|
| OFM19 | 5' GCACTCAAGGCAAGCTTTATTGAGGCTTA 3' | cDNA synthesis |
| 1.3'3'PlCb | 5'ACTACTTAGAGCACTCAAGGCAAGCTTTATTG 3' | Near full-length genome PCR |
| 1.U5Cc | 5'CCTTGAGTGCTCTAAGTAGTGTGTGCCCGTCTGT 3' | Near full-length genome PCR |
| 2.3'3'PlCb | 5'TAGAGCACTCAAGGCAAGCTTTATTGAGGCTTA 3' | Near full-length genome PCR |
| 2.U5Cd | 5'AGTAGTGTGTGCCCGTCTGTTGTGTGACTC 3' | Near full-length genome PCR |
| Vif1 | 5'GGGTTTATTACAGGGACAGCAGAG 3' | 5' Half genome PCR |
| VifR1 | 5'TTCCTCGTCGCTGTCTCCGCTTCTTCCT 3' | 3' Half genome PCR |
| Vif2 | 5'GCAAAACTACTCTGGAAAGGTGAAGGG 3' | 5' Half genome PCR |
| VifR2 | 5'GTCCCCTAGTGGGATGTGTACTTCTGAAYTT 3' | 3' Half genome PCR |

USA) and SMRT sequencing done on the PacBio RSII (Pacific Biosciences Inc) yielding above 20,000 reads of suitable lengths per SMRT library.

## Identification of the transmitted/founder virus

An algorithm described by Dilernia et al [85] that stratifies unique reads from the different genomes and estimates consensus within each genome strata was used to remove sequencing error. All 9kb viral sequences were then aligned in Geneious bioinformatics software (Biomatters, Aukland, NZ) using MUSCLE, followed by hand aligning. Phylogenetic analyses were performed by maximum likelihood parsimony with 100 bootstraps. Pairwise distances for each intra patient variant were extracted using MEGA7 using the Poisson correction model. The Los Alamos National Database HIV Consensus/Ancestral Sequence Alignments were used as reference sequences.

A multi-step approach for resolving the multiplicity of transmission variants by combining previously published methods was applied. First, the presence and number of phylogenetically distinct clusters of recipient sequences arising from distinct source variants using maximum likelihood (ML) trees to categorize distinct sets of identical or nearly identical sequences within the same patient was inferred with bootstrap resampling with 1000 replicas to assess the robustness of the phylogenetic trees. Next, the distribution of pairwise viral diversity and highlighter plots was analyzed. Multivariant infection was inferred from the distribution of hamming distance and confirmed by visual inspection of sequence alignments and variant clustering on phylogenetic trees.

## Cloning of *gag* chimeras

To study the role of the *gag* gene in viral fitness, chimeric molecular clones were generated on a NL4.3 proviral backbone that only differs by the HIV-1 gag gene using a 3-piece modification of the homologous recombination cloning strategy described by Deymier *et al*, [33, 86] and using primers that contained a 15bp overhang of the adjacent cloning piece to allow for homologous recombination in subsequent steps. The 3 pieces were a 1.5kb piece of patient derived T/F *gag* starting from the initiation codon of *gag* and extending 142 nucleotides after the *gag* stop codon and into *pro*, a 6kb fragment derived from NL4.3 spanning from the start of *pol* to the middle of *env* and a 7.4kb pBluescript vector piece containing a fragment of *env*. A total of 100ng DNA was used at the ration of 3:1:1of the 3 pieces. Colonies that tested positive for Gag were further cultured in 5mL of LB broth for 48 hours at room temperature, after which Gag-NL4.3 chimeric DNA was isolated from cultures using the PureYield Plasmid Miniprep System (Promega, USA) and restriction digestion performed with Sph1 enzyme (New England Biolabs, USA) which has a restriction site within *gag* to confirm correct plasmid size and successful insertion of patient derived *gag* gene. Three identical independent clones per patient were chosen for replication assays in order to ensure backbone fidelity during the cloning process.

To generate virus stocks, 1.5 mg of proviral plasmid clones were used to transfect 293T cells (NIH AIDS Reagent Program, USA) with Fugene-HD Transfection reagent (Promega, USA) and cell supernatants collected 48 hours after transfection and clarified by centrifugation. Virus stocks were titered on the Tzm-bl reporter cell line (NIH AIDS Reagent Program) according to standard protocols.

## In vitro replicative capacity assay

In order to assess the in vitro replicative capacity of the generated infectious molecular clones, GXR25 cells (donation from Eric Hunter's Lab, Emory University, USA) were infected at a

multiplicity of infection (MOI) of 0.01 and 100 μl of viral supernatants collected at 2-day intervals as previously described [87]. Briefly, GXR25 cells and virus were incubated with 5 mg/ml polybrene at 37˚C for 3 hours, washed 5 times with complete R10 media and plated into 24-well plates. Cells were split 1:2 every 2 days, replaced with an equal volume of fresh media, and viral supernatants were taken at days 2, 4, 6 and 8 as previously described. Virion production was quantified using a colorimetric reverse transcriptase assay using the 15-hour protocol (Roche, Switzerland). Based on values obtained for days 2–8, the optimal window for logarithmic growth for most viruses was determined to be between days 2 and 6, as by day 8 many high replicating viruses had exhausted target cells causing a flattening or decline of the replication curve. Therefore, log10-transformed slopes were calculated based on days 2, 4 and 6 for all viruses. Replication scores were generated by dividing the log10-transformed slope of the replication curve for each Gag-NL4.3 chimera by the log10- transformed slope of wild-type NL4.3. Three independent Gag-NL4.3 chimera clones per patient recipient were run and their replicative capacity scores averaged.

### Flow cytometry analysis

Cryopreserved PBMCs isolated at 1, 9- and 30-months post-infection were analysed by flow cytometry using the panels detailed in **S2 and S3 Tables**. Frozen PBMCs were thawed, washed once with R20 (RPMI 1640 (Sigma) containing 20% FBS (Sigma), 1% 1 M Hepes buffer (Sigma), 1% L-glutamine (Sigma), 1% penicillin–streptomycin (Sigma), and 1% sodium pyruvate (Sigma)). 1 million cells were stained for 20 min at room temperature with Aqua Live/Dead amine dye-AmCyan (Invitrogen). Cells were then washed once with R10 and anti-CCR7-APCR700 was added to the cells and incubated at 37˚ C for 15 min. The rest of the monoclonal antibodies, except anti-Ki67, were added and incubated at room temperature for 20 min. After washing with PBS, Cytofix/Cytoperm (BD) was added to cells and incubated at 4˚C for 20 min. After washing with Perm wash buffer, anti-Ki67 antibody was added to the cells. The 12-parameter panel is shown in **S2 Table**. At least 30,000 viable $CD3^+$ lymphocytes were acquired on the Fortessa flow cytometer using FACSDiva software (BD Biosciences). Analysis was done on FlowJo V10.1.1 (Treestar Inc, USA). To correct for spectral overlap between fluorochromes, compensation controls for each antibody were prepared by staining anti mouse antibody $BD^{TM}$ Compensation Particles (BD, UK) and negative control particles (BD Biosciences) with the titrated antibody volumes. To gate for the different T-cell subsets, $CD3^+$ singlet viable lymphocytes were gated for CD4 and CD8 expression, and single gates or CD38, HLA-DR, CD27, CD45RO, CCR7, CD57, PD-1, Ki67, Granzyme B, Perforin and CD107a carried out. Boolean gates were then created for the populations of interest, and the expression profiles of the different cells analyzed on GraphPad Prism V7 (San Diego, USA), Pestle V1.7 (Maryland, USA), Spice V6 (Maryland, USA) and JMP V14 (Marlow, UK). B cells were defined as $CD3^-$ $CD19^+$ lymphocytes and further characterized for expression of CD21, CD27, CD10, CD38, IgM, IgD, CD86 and PD-L1.

### Statistical analysis

Statistical analysis and graphical presentation were performed using GraphPad Prism software version 7 and JMP statistical software version 14 (SAS, USA). Mann-Whitney U test was done for nonparametric unpaired observations or the Kruskal-Wallis test when multiple sets of nonparametric unpaired observations were compared. The Wilcoxon matched pairs test was used to compare statistical difference between two sets of paired nonparametric data. All bivariate continuous correlations were performed using standard linear regression. One-way comparison of means was performed using Student's t test with Benjamini-Hochberg correction for

multiple comparisons, and one-tailed p values are reported. Kaplan–Meier survival curves were performed using an endpoint defined as a single $CD4^+$ T-cell count reading less than 350cells/mm$^3$, and statistics reported for survival analyses were generated from the log-rank test. P values of $<0.05$ were of statistical significance.

## Supporting information

**S1 Fig.** Phylogenetic tree based on A) 3 prime half genome nucleotide sequences from 13 participants. Sequences on branches of the same colour were derived from the same individual, while sequences in black correspond to subtype references.
(TIF)

**S2 Fig. Phylogenetic tree based on 5 prime half genome nucleotide sequences from 38 participants.** Sequences on branches of the same colour were derived from the same individual, while sequences in black correspond to subtype references.
(TIF)

**S1 Table. Cohort characteristics.**
(TIF)

**S2 Table. T cell phenotyping panel comprising of markers of activation, differentiation and exhaustion of $CD4^+$ and CD8 T$^+$ cells.**
(TIF)

**S3 Table. B cell phenotype panel to determine whether viral replicative capacity influences B cell maturation, differentiation and class switching.**
(TIF)

## Acknowledgments

We thank all the volunteers in KEMRI-CGMRC who participated in this study and all the staff at the KEMRI-CGMRC HIV Research Project in Kilifi and Mtwapa who made this study possible. We thank Brian Abel, Jonathan Hare and the VISTA project group for important scientific output. We also thank Elizabeth Wahome, Caroline Ngetsa, John Brennan, Brendan McAtarsney, Jon Allen, Sheng Luo and Paul Farmer for technical assistance, sample management, and database management.

## Author Contributions

**Conceptualization:** Gladys N. Macharia, Matthew A. Price, Eduard J. Sanders, Eric Hunter, Jill Gilmour.

**Data curation:** Gladys N. Macharia, Ecco Staller, Dario Dilernia, Daniel Wilkins.

**Formal analysis:** Gladys N. Macharia, Ecco Staller, Dario Dilernia, Daniel Wilkins.

**Funding acquisition:** Pat Fast, Matthew A. Price, Eduard J. Sanders, Eric Hunter, Jill Gilmour.

**Investigation:** Gladys N. Macharia, Ecco Staller, Dario Dilernia, Daniel Wilkins, Heeyah Song, Eric Hunter.

**Methodology:** Gladys N. Macharia, Ling Yue, Ecco Staller, Dario Dilernia, Heeyah Song, Nesrina Imami, Eric Hunter, Jill Gilmour.

**Project administration:** Gladys N. Macharia, Eduard J. Sanders, Eric Hunter, Jill Gilmour.

**Resources:** Gladys N. Macharia, Nesrina Imami, Eduard J. Sanders, Eric Hunter, Jill Gilmour.

**Supervision:** Edward McGowan, Deborah King, Nesrina Imami, Eric Hunter, Jill Gilmour.

**Validation:** Gladys N. Macharia.

**Visualization:** Gladys N. Macharia.

**Writing – original draft:** Gladys N. Macharia.

**Writing – review & editing:** Gladys N. Macharia, Deborah King, Nesrina Imami, Matthew A. Price, Eduard J. Sanders, Eric Hunter, Jill Gilmour.

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
