## [Decision Letter · Decision Letter 0]

27 May 2020

Dear Dr Macharia,

Thank you very much for submitting your manuscript "Infection with multiple HIV-1 founder variants is associated with lower viral replicative capacity, faster CD4+ T cell decline and increased immune activation during acute infection" for consideration at PLOS Pathogens. As with all papers reviewed by the journal, your manuscript was reviewed by members of the editorial board and by several independent reviewers. In light of the reviews (below this email), we would like to invite the resubmission of a significantly-revised version that takes into account the reviewers' comments.  Both reviewers raise relevant points.  The issues of independence in the statistical analyses raised by Reviewer 1 need to be carefully considered as they may negatively impact the quality of your interpretations.

We cannot make any decision about publication until we have seen the revised manuscript and your response to the reviewers' comments. Your revised manuscript is also likely to be sent to reviewers for further evaluation.

Sincerely,

Ronald Swanstrom

Associate Editor

PLOS Pathogens

Richard Koup

Section Editor

PLOS Pathogens

Kasturi Haldar

Editor-in-Chief

PLOS Pathogens

orcid.org/0000-0001-5065-158X

Michael Malim

Editor-in-Chief

PLOS Pathogens

orcid.org/0000-0002-7699-2064

Reviewer's Responses to Questions

**Part I - Summary**

Reviewer #1: In this manuscript, investigators explore number of variants during MSM HIV-1 transmission and possible implications regarding the fitness of the transmitted variants. They examined 38 MSM participants relatively early after infection. They isolated nearly full length genomes and characterized 15 of 38 individuals having multiple variants.

In general, this manuscript is well written and presents the data well.

There are four main findings and associated concerns and criticisms with their observations / conclusions.

1) A larger percentage of acutely infected MSM individuals harbor multiple founder strains. This is not a novel finding. Indeed the authors cite the publication from Li et. al. although they fail to adequately acknowledge that this is not a new finding. Authors should make a more persuasive argument why there finding is unique.

2) They demonstrate that individuals with multiple as compared to single founders generally faster disease progression. They found that multiple as compared to single had faster CD4 decline although not higher virus level. This finding is also not novel, and it has also been previously demonstrated in different cohorts.

3) They examined the replication capacity conferred by the gag gene among the multi- and single founder cases. In figure 3, they find that among the multiple founder cases, the minority strain’s gag often confers higher replicative capacity. Fig. 3B appears to hint at this finding although, it has not been adequately analyzed. I would suggest that they do a matched pairwise comparison between the major variant and an average or median of the minority variants among the 7 analyzed cases. It is also unclear as to why the authors only analyzed 7 of the 15 multivariant cases. The data would be stronger with a larger number of cases. Fig. 3C is also not accurate. The data points for the correlation are not independent (numerous data points are from the same individuals). An assumption for the statistical test (Spearman rank correlation) is that all data points are independent.

4) Fig. 4 requires greater explanation. Which of the gag sequences were used in the multiple variants. Did it comprise only 1 variant from 12 individuals with multiple founders? If so, how and why was this 1 variant chosen? Are there multiple gag sequences from the same patients for the multiple variants column in this figure? If so, this would violate the assumption of independence for the statistical test. Should a weighted average (based on frequency of detection) of gag fitness from the individuals with multiple variants be used?

5) They also examined changes in T and B cells among individuals with multiple and single variants. This analysis is incomplete. There are only a limited number of individuals examined especially for those with multiple founders. Why was the analysis restricted to a small number of individuals? There are also multiple comparisons, and authors should acknowledge the possibility of the type 1 error due to examination of limited number of specific samples and multiple comparisons.

Reviewer #2: The authors investigate the role of HIV-1 multivariant infection on disease progression in an MSM population. Furthermore Gag-mediated replication capacity is investigated as one factor influencing multiplicity of infection and disease outcomes. The information is presented in a straightforward and clear manner, with good linkage to key related studies.

While both the aspect of a reduced bottleneck in MSM versus heterosexual transmission, as well as an association of multiplicity of infection with disease progression have previously been reported on, this study is novel in that it investigates replication capacities of HIV variants from single versus multivariant infection and reports a more rapid decline in CD4+ T cells, a known marker of disease progression, in individuals infected with more than one variant. In addition, the authors conclusions reiterate the need for treatment of existing STIs as a tool in the prevention and slowing of disease in vulnerable populations.

Many factors, both viral and host, may influence the progression of clinical disease, as is pointed out by the authors themselves. There are therefore certain areas in the manuscript that would benefit from greater context or supporting information to strengthen the relationships identified.

**Part II – Major Issues: Key Experiments Required for Acceptance**

Reviewer #1: (No Response)

Reviewer #2: (No Response)

**Part III – Minor Issues: Editorial and Data Presentation Modifications**

Reviewer #1: (No Response)

Reviewer #2: 1. The authors mention high risk behavior and presence of STIs within the cohort. In addition, while epidermal barrier is primarily associated with increased risk of multivariant infection in MSM, presence of an existing STI is also noted as a possible determinant. Could the authors address the following:

a. Can the authors comment on the presence of STIs potentially having an influence on the state of immune cell markers and differences in characteristics of CD4 T cell and B cell characteristics between the two groups? Can the authors account for presence or absence of an STI (or types of STIs if prevalent in both groups) and comment on this as a potential confounder, if this information exists?

b. Are data available pre-infection for a baseline indication of immune cell markers prior to HIV infection between those who acquired single versus multivariant infection?

c. Could the authors comment as to whether any STI treatment was provided post HIV diagnosis and if so could this potentially contribute to the observed improvement in immune perturbances?

2. CTL escape can occur rapidly in acute infection and have particular impact on viral fitness when occurring in Gag. Can the authors comment as to a potential role of CTL escape in Gag in the difference in replication capacity of dominant versus minor variants in the multivariant infections. Were putative fitness-associated Gag mutations screened for in the clones utilized for the replication assays?

3. Can the screening intervals for HIV infection be indicated in the methods?

4. The authors introduce the relevance of the study as linked to prevention and cure of HIV, and later in the discussion section place emphasis on need for understanding pathogenesis. While all may apply, it would be good to provide a clearer description in terms of rationale and where this study specifically fits in.

5. In the discussion, the reduced bottleneck in MSM transmission is reported as though novel, while the authors do reference previous studies on this phenomenon. It would be of greater interest to compare the percentage single vs multi-variant infection in this study to that of previous studies, or the number of participants investigated, and to comment on any contrasts and factors that may have influenced differences between MSM cohort studies, if any.

PLOS authors have the option to publish the peer review history of their article (what does this mean?). If published, this will include your full peer review and any attached files.

Reviewer #1: No

Reviewer #2: No
---

## [Editor Report · Decision Letter 1]

3 Aug 2020

Dear Dr Macharia,

We are pleased to inform you that your manuscript 'Infection with multiple HIV-1 founder variants is associated with lower viral replicative capacity, faster CD4+ T cell decline and increased immune activation during acute infection' has been provisionally accepted for publication in PLOS Pathogens.

Best regards,

Ronald Swanstrom

Associate Editor

PLOS Pathogens

Richard Koup

Section Editor

PLOS Pathogens

Kasturi Haldar

Editor-in-Chief

PLOS Pathogens

orcid.org/0000-0001-5065-158X

Michael Malim

Editor-in-Chief

PLOS Pathogens

orcid.org/0000-0002-7699-2064
---

## [Editor Report · Acceptance letter]

27 Aug 2020

Dear Dr Macharia,

We are delighted to inform you that your manuscript, "Infection with multiple HIV-1 founder variants is associated with lower viral replicative capacity, faster CD4+ T cell decline and increased immune activation during acute infection," has been formally accepted for publication in PLOS Pathogens.

Best regards,

Kasturi Haldar

Editor-in-Chief

PLOS Pathogens

orcid.org/0000-0001-5065-158X

Michael Malim

Editor-in-Chief

PLOS Pathogens

orcid.org/0000-0002-7699-2064